# Sleep Disturbances Across Dementias and Cognitive Decline: Study Protocol for a Systematic Review and Network Meta-Analysis of Polysomnographic Findings

**DOI:** 10.3390/jcm14207437

**Published:** 2025-10-21

**Authors:** Annibale Antonioni, Arianna Della Valle, Caterina Leitner, Emanuela Maria Raho, Edward Cesnik, Jay Guido Capone, Maria Elena Flacco, Francesca Casoni, Paola Proserpio, Luigi Ferini-Strambi, Andrea Galbiati

**Affiliations:** 1Department of Neuroscience and Rehabilitation, University of Ferrara, 44121 Ferrara, Italy; 2Department of Clinical Neurosciences, Neurology-Sleep Disorders Center, IRCSS San Raffaele Institute, 20127 Milan, Italy; 3Faculty of Psychology, “Vita-Salute” San Raffaele University, 20127 Milan, Italy; 4Department of Neuroscience, Ferrara University Hospital, 44124 Ferrara, Italy; 5Department of Environmental and Prevention Sciences, University of Ferrara, 44121 Ferrara, Italy

**Keywords:** dementia, mild cognitive impairment (MCI), neurodegenerative disorders, polysomnography (PSG), sleep, subjective cognitive impairment (SCI)

## Abstract

Sleep disturbances are increasingly recognized as early and clinically meaningful features in the pathophysiology of neurodegenerative diseases. Polysomnography (PSG), i.e., the gold standard for objectively characterizing sleep architecture, may provide non-invasive and scalable biomarkers for both early detection and differential diagnosis of dementia. This systematic review and network meta-analysis aims to synthesize existing evidence on PSG-derived sleep alterations across the neurodegenerative continuum, including subjective cognitive impairment, mild cognitive impairment, Alzheimer’s disease, frontotemporal dementia, Lewy body dementia, and Parkinson’s disease dementia, compared to healthy controls. It will adhere to the Preferred Reporting Items for Systematic Reviews and Meta-Analysis (PRISMA) guidelines for reporting systematic reviews that include network meta-analyses, and it has been registered with the International Prospective Register of Systematic Reviews (PROSPERO) under registration number CRD420251114418. We will include peer-reviewed studies with nocturnal PSG data from adult participants, categorized using validated diagnostic criteria, and scored according to the most recent American Academy of Sleep Medicine guidelines. Both pairwise and network meta-analyses will be conducted using standardized mean differences to quantify group-level effects. Additional analyses will explore the correlations between PSG parameters, severity of cognitive impairment, behavioral symptoms, treatments, and relevant comorbidities. Longitudinal data, where available, will be analyzed separately to evaluate prognostic value. This study will provide a comprehensive synthesis of PSG alterations across neurodegenerative disorders, offering insights into their diagnostic utility and potential as early markers for stratification in clinical trials of disease-modifying therapies.

## 1. Introduction

Primary neurodegenerative dementias encompass a diverse group of chronic conditions marked by a gradual decline in cognitive functions, ultimately leading to loss of independence in daily activities and, eventually, death [1,2,3]. This heterogeneous group includes Alzheimer’s disease (AD), frontotemporal dementia (FTD), Lewy body dementia (LBD), and Parkinson’s disease dementia (PDD), each distinguished by specific neuropathological hallmarks and anatomical patterns of brain involvement, which result in diverse clinical manifestations [4,5,6]. Considering the substantial healthcare, societal, and economic burden associated with these conditions, extensive research efforts are focused not only on identifying biomarkers that can predict disease progression but also on enabling accurate differential diagnosis among the various subtypes, especially in the earliest stages, when clinical features often overlap [7,8,9,10]. This is particularly important, as emerging disease-modifying treatments appear to be most effective when initiated early, before extensive neurodegeneration has occurred [11,12]. Moreover, since these new therapies are often designed to target specific pathological hallmarks, such as beta-amyloid or phosphorylated tau in the case of AD, early and precise phenotyping is crucial to ensure a proper treatment strategy [13,14,15].

Importantly, it is now well-established that patients with mild cognitive impairment (MCI) often represent a prodromal stage of neurodegenerative dementia and may therefore have a critical window for therapeutic intervention [16,17]. More recently, evidence suggests that even some instances of subjective cognitive impairment (SCI) might represent the earliest detectable stage in the neurodegenerative continuum, particularly in the context of AD [18,19]. This underscores the pressing need to develop accurate, accessible, and clinically viable biomarkers that can reliably identify patients eligible for early treatment. However, currently, diagnosis confirmation often depends on invasive and resource-intensive methods, such as positron emission tomography (PET) with specific radiotracers, advanced magnetic resonance imaging (MRI) techniques, or the detection of neuropathological markers in cerebrospinal fluid (CSF) [8,20,21,22,23]. Furthermore, promising neurophysiological approaches, such as high-density electroencephalography (HD-EEG) recordings, require advanced acquisition systems and complex analytical pipelines, making them difficult to implement in routine clinical settings [24,25]. While highly informative, these approaches are expensive, not easily scalable to the general population, and require dedicated time and expertise [26].

For these reasons, as emphasized by recent biological diagnostic criteria for AD, there is growing interest in more scalable biomarkers, such as blood-based assays or indices derived from non-invasive brain stimulation techniques [27,28,29,30,31]. However, their use remains mostly limited to research settings, and thorough real-world validation will be necessary before they can be incorporated into standard clinical workflows. In this context, exploring sleep physiology and its alterations offers a promising path.

Sleep is traditionally defined as a recurring physiological state characterized by a reversible decrease in consciousness, voluntary movements, and the ability to interact with the environment [32]. Despite decades of extensive research, its complete range of functions remains only partially understood. Nevertheless, a growing body of evidence highlights its crucial role in a wide range of processes, including synaptic homeostasis, energy metabolism, emotional regulation, and cognition [33,34,35]. In addition, recent findings have highlighted sleep as a critical period for clearing neurotoxic waste products from the central nervous system (CNS) through mechanisms such as glymphatic drainage [36,37]. These processes are crucial for maintaining neural integrity and function and are notably disrupted in neurodegenerative diseases, which are typically characterized by the accumulation of misfolded proteins, synaptic dysfunction, and impaired neuroplasticity and energy metabolism, all of which underlie the neurobiological basis of the clinical symptoms observed [38,39,40,41]. These considerations become even more relevant when acknowledging that neurodegenerative conditions primarily affect the elderly population, in which sleep architecture is already altered due to age-related changes, and often further exacerbated by comorbidities and widespread use of polypharmacy, both of which may critically impair sleep quality and continuity [42,43,44,45].

Therefore, it is unsurprising that sleep disturbance, both in Non-Rapid Eye Movement (NREM) sleep, which is characterized by progressive cortical slowing and synchronization, and REM stages, which is marked by a drop in muscular tone, faster brain rhythms, and characteristic eye movements, is a consistent feature in patients with neurodegenerative disorders [46,47,48,49]. Notably, the relationship appears to be bidirectional: sleep alterations are often present even in the early or prodromal stages of the neurodegenerative continuum (i.e., SCI and MCI) and are associated with a higher risk of progression toward overt cognitive decline [50,51,52,53]. Indeed, the more severe the sleep disruption, the greater the risk of future dementia [54,55]. Similarly, more serious clinical symptoms are usually associated with greater changes in sleep patterns [56,57]. Furthermore, the link between REM Sleep Behavior Disorder (RBD) and underlying neuropathological changes, particularly within the spectrum of alpha-synucleinopathies, is now well established [58,59]. Consistently, increasing evidence suggests that RBD may precede the overt clinical onset of these disorders by several decades, supporting its recognition as a legitimate prodromal stage of neurodegenerative disease. Notably, while the strongest links have been demonstrated in synuclein-related conditions, emerging data indicate that RBD may also represent an early marker of vulnerability in other neurodegenerative contexts as well [60].

Importantly, polysomnography (PSG), which represents the gold standard for objective sleep assessment, has sometimes revealed peculiar patterns of sleep disruption across different dementia subtypes, suggesting the potential for a sleep-based signature of disease [57,61,62,63,64]. In particular, changes in key features of physiological NREM sleep, such as reductions in slow-wave activity and alterations in spindle density and morphology, have been consistently reported across various neurodegenerative conditions, reflecting the early dysfunction of thalamocortical and cortical networks that are critically involved in memory consolidation and synaptic homeostasis [65,66]. Moreover, it is worth noting that PSG provides several key strengths: it is non-invasive, repeatable, and capable of characterizing sleep architecture with high precision, making it a potentially valuable alternative to more invasive and costly diagnostic tools, such as PET or CSF analysis [67]. Indeed, repeatedly administering the test over time offers considerable potential for tracking disease progression, as demonstrated in various pathological settings [68,69,70]. Taken together, these findings suggest that PSG-derived sleep metrics may serve as a valuable tool not only for distinguishing among disease subtypes but also for monitoring individual disease progression in neurodegenerative disorders [50,71,72,73]. This could have profound implications not only for advancing our comprehension of disease pathophysiology, even in atypical cases of neurodegeneration, where mechanisms remain only partially understood since each element of sleep is intricately regulated by complex neural circuits that are disrupted by neuropathological processes, but also for the earlier identification of individuals at risk [74,75,76,77]. This would enable timely intervention with disease-modifying therapies that are ideally tailored to specific neurodegenerative profiles, including both pharmacological and non-pharmacological interventions [78,79,80].

However, to the best of our knowledge, no comprehensive quantitative synthesis of PSG data across the entire spectrum of cognitive decline is currently available, nor have existing studies systematically compared PSG parameters across diagnostic categories or mapped their longitudinal changes compared to healthy subjects (HS). Therefore, we aim to conduct a systematic review and network meta-analysis to address this important gap in the literature and provide essential evidence to enhance our understanding of sleep changes across the neurodegenerative spectrum. Indeed, sleep-based assessments may provide non-invasive, physiologically grounded markers to improve early detection, support differential diagnosis, and ultimately guide therapeutic decision-making in individuals along the neurodegenerative spectrum.

## 2. Materials and Methods

This systematic review and network meta-analysis will adhere to the Preferred Reporting Items for Systematic Reviews and Meta-Analysis (PRISMA) guidelines for reporting systematic reviews that include network meta-analyses (PRISMA checklist in Appendix A) and has been registered with the International Prospective Register of Systematic Reviews (PROSPERO) under registration number CRD420251114418 [81,82]. Our primary review question will be “What are the differences in PSG sleep profiles among individuals with neurodegenerative cognitive decline?” Specifically, the entire spectrum of neurodegenerative and prodromal cognitive decline, including SCI, MCI, AD, FTD, LBD, and PDD, will be considered.

### 2.1. Eligibility Criteria

#### 2.1.1. Types of Studies

We will include randomized controlled clinical trials (RCTs), cohort studies, case–control studies, and cross-sectional studies, either published as primary analyses or as sub-analyses of larger population-based cohort studies. Studies published in English will be included. Furthermore, studies must meet the following criteria to be eligible:Regarding patient categorization, currently accepted international clinical diagnostic criteria must be used [5,83,84,85,86,87,88,89].PSG scoring must apply the most recent scoring criteria from the American Academy of Sleep Medicine (AASM) available at the time of the study. The use of automated scoring algorithms may also be considered, provided they are validated against the AASM standard.Studies must report PSG data from one or more groups of patients diagnosed with cognitive decline, either compared with each other or versus HS.PSG parameters must be reported as mean and standard deviation (SD) or median and interquartile range (IQR, expressed as Q1–Q3).PSG sleep stage parameters must be expressed as percentages, thus corrected for total sleep time (TST). When raw data in minutes and TST are provided, the data will be manually corrected.

Preclinical studies (e.g., on cellular and animal models), conference abstracts, and literature reviews will be excluded.

#### 2.1.2. Types of Populations

Adult individuals affected by neurodegenerative cognitive impairments, i.e., with a diagnosis of SCI, MCI, AD, FTD, LBD, or PDD, formulated according to internationally recognized clinical criteria, will be considered [5,83,84,85,86,87,88,89].

#### 2.1.3. Types of Interventions

The goal of this work is not to assess the effects of specific pharmacological or non-pharmacological treatments on PSG parameters in patients across the cognitive decline spectrum. However, when available, we will differentiate between baseline PSG profiles and those recorded at intermediate or post-treatment time points to account for potential neuropharmacological and/or neurophysiological mechanisms associated with specific interventions. Additionally, baseline stratification, where feasible, will take into account whether participants were undergoing treatments known to affect sleep, versus those not receiving such therapies.

#### 2.1.4. Types of Comparators

HS PSG data will be used as a common comparator to indirectly evaluate differences between patient groups. Additionally, when possible, PSG parameters will be compared within individuals on the same spectrum but with different degrees of severity, such as HS, SCI, MCI due to AD, and overt AD, to outline a longitudinal profile of sleep changes in the same neuropathological continuum.

#### 2.1.5. Types of Outcomes

Among the eligible studies, the main goal is to analyze PSG nocturnal parameters for each group, including HS, SCI, MCI, AD, FTD, LBD, and PDD. Moreover, we will also examine characteristic sleep graphoelements (e.g., slow waves, spindles, K-complexes) whenever they are reported. Furthermore, if feasible, we will also assess additional outcomes. Specifically, correlations will be examined between the severity of cognitive impairment, as measured by standardized tools—including both general cognitive tests like the Mini-Mental State Examination (MMSE) and the Montreal Cognitive Assessment (MoCA)—as well as disease-specific assessments such as the Alzheimer’s Disease Assessment Scale–Cognitive Subscale (ADAS-Cog), and key nocturnal PSG parameters [90,91,92]. Similarly, correlations between behavioral and neuropsychiatric disturbances, evaluated using validated tools such as the Neuropsychiatric Inventory (NPI), and nocturnal sleep PSG data will be explored [93]. As previously noted, both pharmacological and non-pharmacological interventions will be considered for stratification at baseline and, when longitudinal data are available, to assess potential treatment-related changes in sleep architecture. Furthermore, the presence of comorbid conditions (and related treatments) known to affect sleep architecture, such as diabetes, chronic pain syndromes, or other relevant medical issues, will be documented whenever possible to evaluate their potential impact on the outcomes of interest [94,95]. In addition, in conditions suggestive of prodromal stages, such as SCI and MCI, available longitudinal data will be utilized to develop predictive models of disease progression based on baseline PSG findings. Finally, when available, we will also extract quantitative electroencephalography (qEEG) metrics, which involve the quantitative analysis of EEG signals using advanced mathematical and statistical methods, to better characterize sleep features [96,97,98].

### 2.2. Search Strategy

A bibliographic search will be performed across different databases (MedLine, Scopus, EBSCO, and Web of Science), with no date restriction. Various combinations of the following search terms related to eight main domains will be used, after adjustment for each database: “polysomnografy OR PSG”(title/abstract) AND “sleep”(title/abstract) AND “Alzheimer’s Disease OR AD”(title/abstract) or “subjective cognitive impairment OR SCI” (title/abstract) or “mild cognitive impairment OR MCI”(title/abstract) or “fronto-temporal dementia OR FTD”(title/abstract) or “Lewy body dementia OR LBD”(title/abstract) or “Parkinson’s disease dementia OR PDD”(title/abstract). The reference lists of reviews and retrieved articles will also be searched for additional pertinent papers.

### 2.3. Study Screening and Selection

In the first phase, specific search strings will be formulated for each queried database, and the retrieved studies will be listed in Rayyan (https://www.rayyan.ai/, accessed on 9 October 2025). Titles and abstracts will be blinded against inclusion and exclusion criteria by two independent reviewers, and duplicate studies will be removed. If a disagreement arises between the two reviewers, a third research team member will be contacted to resolve the issue regarding the article. In the second phase, the full texts of the articles selected in the first phase will be retrieved, and the eligibility criteria will be applied to them. A PRISMA flow diagram showing the number of studies included and excluded at each stage of the study selection process will be provided (Appendix A) [99].

### 2.4. Data Extraction

The data extracted will include publication details, such as authors, publication year, title, journal, database sample size, diagnostic category(ies), and sample characteristics for each group, including sample size, percentage of females, average age, and, where reported, relevant genetic factors in the study sample (e.g., Presenilin-1/2 and apolipoprotein E in the context of AD, chromosome 9 open reading frame 72 in FTD) [5,100]. Moreover, following the same methodological approach of previous works from our group, specific PSG parameters, such as NREM sleep stages 1 (N1, %), 2 (N2,%), 3 (N3, %), REM sleep duration (%), Sleep Onset Latency (SOL, minutes), TST (minutes), time in bed (TIB, minutes), Sleep Efficiency (SE, %), Wake After Sleep Onset (WASO, minutes), awakenings and movements (AM) index (number of events per hour of sleep), number of awakenings (NA), number of arousals (Arousal Index, AI), REM latency (minutes), and REMs density for each included group will be evaluated [101]. In addition, disease characteristics, assessed by both general and disease-specific cognitive tests, behavioral and neuropsychiatric tests, comorbidities, and pharmacological and non-pharmacological treatments, will be taken into consideration. Moreover, when available, qEEG metrics, including spectral power across standard frequency bands, such as delta (0.5–4 Hz), theta (4–8 Hz), alpha (8–13 Hz), sigma (12–16 Hz), and beta (13–30 Hz), along with related ratios (e.g., delta-alpha ratio, delta-theta ratio), will be considered. Finally, whenever reported, we will also consider characteristic sleep graphoelements (e.g., spindles, K-complexes), given their well-established relevance to sleep physiology and dementia research.

### 2.5. Quality Assessment

Two independent authors will use standardized scales to evaluate the quality of the included studies based on their design, such as Version 2 of the Cochrane risk-of-bias tool for randomized trials (RoB 2) for RCTs, and the Newcastle-Ottawa Quality Assessment Scale (NOS) for observational studies [102,103]. Furthermore, for each meta-analysis with more than 10 publications, publication bias will be evaluated both visually using funnel plots and statistically through Egger’s regression asymmetry test [104]. The strength of the body of evidence will be assessed through the Grading of Recommendations, Assessment, Development, and Evaluations (GRADE) certainty ratings. Using GRADE guidelines [105,106], the researchers will examine the following domains for a decrease in certainty: risk of bias, imprecision of true effects, inconsistency of effects, indirectness of outcomes, and publication bias.

### 2.6. Data Synthesis

A qualitative summary of the data will first be presented, including sample size, diagnostic categories, percentage of females, and average age. Additionally, PSG parameters, including NREM sleep stages, REM sleep duration, SOL, TST, TIB, SE, WASO, AM index, NA, REM latency, REMs density, qEEG metrics, as well as scores in general and disease-specific cognitive, behavioral, and neuropsychiatric tests, along with comorbidities and both pharmacological and non-pharmacological treatments, will be summarized. The network’s geometry will be based on pairwise comparisons derived from existing literature on PSG parameters in patients with neurodegenerative cognitive disorders. Most network connections will involve edges linking neurodegenerative conditions (i.e., SCI, MCI, AD, FTD, LBD, and PDD) to HS, which acts as a common comparator. Often, direct comparisons between different disease groups (e.g., MCI vs. AD, or FTD vs. LBD) may not be available. In those cases, indirect comparisons will be made through their shared connection with the HS group. The edges of the network will be weighted by the number of studies contributing to each comparison. At the same time, nodes will represent diagnostic groups and be scaled according to the sample size of each included population. In cases where studies include more than two diagnostic groups (e.g., HS, MCI, AD), triangular connections (three-arm studies) will enable more complex network structures and allow for indirect estimation across multiple conditions. Therefore, the analysis will be carried out in two phases. First, a series of pairwise meta-analyses will be performed to examine differences in PSG parameters between HS and each neurodegenerative condition separately (e.g., HS vs. MCI, HS vs. AD, etc.). Standardized mean differences (SMD) will be used as the summary effect size measure to indicate comparisons between groups. In case some of the included studies express PSG parameters as medians and ranges, we will use the method described by Hozo et al. to obtain the corresponding means and SDs [107]. In case interquartile ranges (IQR) rather than ranges are reported, they will be divided by 1.35 to obtain the equivalent SD [108]. For this phase, we will use a fixed-effects model, supported by the expected standardized nature of PSG parameters in the literature. When relevant, disease severity and treatments, as well as comorbidities (and related therapies) that can alter sleep patterns, will be included as moderators in meta-regression models to examine their potential effect on sleep architecture. In the second phase, a network meta-analysis will be conducted to enable both direct and indirect comparisons between diagnostic categories, particularly where head-to-head data are sparse. Due to the anticipated clinical heterogeneity, we will use a random-effects model for the network meta-analysis. RevMan 5.3 (The Cochrane Collaboration, 2014) and Stata, version 13.1 (Stata Corp. College Station, TX, USA: 2013) will be used to analyze the data.

### 2.7. Analysis of Subgroups or Subsets

Heterogeneity across studies will be assessed using the I^2^ statistic in both pairwise and network analyses. For those PSG parameters supported by the three-arm design, we will determine local inconsistency using the Separate Indirect from Direct Evidence (SIDE) method, which compares direct and indirect estimates within the same triad of groups [109]. However, for all parameters included, global inconsistency across the entire network will be evaluated using the Q statistic for inconsistency [110]. Additionally, if sufficient data will be available, all analyses will be stratified by several demographic and clinical characteristics: (a) gender; (b) clinical status (presence vs. absence of comorbidities); (c) type of treatment (pharmacological; non-pharmacological; no treatment). Moreover, if more than 10 studies are available, a meta-regression will be performed to explore the association between the selected covariates and the outcome of interest.

### 2.8. Sensitivity Analysis

Sensitivity analyses will be conducted to examine the influence of outliers (via Galbraith plots where I^2^ ≥ 40%), influential studies (via leave-one-out meta-analysis), and studies with a high risk of bias.

## 3. Discussion

Given the high diagnostic complexity of neurodegenerative dementias, especially in their early stages, it is crucial to develop tools that enhance diagnostic accuracy, particularly methods that are non-invasive, affordable, and reproducible [111,112,113,114]. Such approaches would facilitate the timely enrollment of patients in clinical trials for emerging disease-modifying therapies when these interventions are most likely to be effective. This issue is especially important today, as the aging population is likely to significantly raise the prevalence of dementia [115]. Furthermore, the recent SARS-CoV-2 pandemic has underscored significant associations with an elevated risk of subsequent cognitive decline [116]. These associations appear both direct—through potential CNS injury mediated by multiple mechanisms—and indirect, as the infection has been shown to increase the incidence of conditions such as delirium, which is defined as an acute and fluctuating disturbance of attention, awareness, and cognition [117,118]. Notably, delirium is a well-established predictor of long-term cognitive decline and dementia [119]. Therefore, given the global scale of the pandemic, an increase in cognitive impairment is expected in the coming years, emphasizing the need for reliable tools to distinguish between different causes of decline [116,120]. Sleep analysis, which is increasingly recognized as critically involved in the pathophysiology of neurodegenerative disorders, offers a unique insight into disease mechanisms [121]. Thus, it is hoped that this work will reveal whether specific PSG signatures differ systematically across dementia syndromes. Such findings would provide essential insights into the pathophysiological mechanisms linking neurodegeneration to sleep disruption, such as highlighting the selective vulnerability of thalamocortical or brainstem circuits in different dementia subtypes [10,122]. Importantly, by combining sleep features with current neurodegeneration models, we can explore whether specific PSG features represent downstream markers of widespread network issues or early factors that drive disease progression. This could not only enhance our understanding of disease-specific changes, potentially aiding in the differential diagnosis of particular dementia subtypes, but it may also provide clinically useful biomarkers to guide personalized treatment strategies. Indeed, it is well established that neuromodulatory interventions can be tailored to individual neurophysiological profiles, and it seems reasonable that specific sleep-related alterations may serve as novel therapeutic targets for these techniques [123,124,125]. This approach could foster the integration of sleep metrics into precision medicine frameworks, paving the way for early detection, patient-tailored management, and the development of novel therapeutic targets. Notably, PSG-derived biomarkers may hold prognostic value, enabling clinicians to stratify patients based on their risk of rapid cognitive decline or responsiveness to targeted interventions. Therefore, we believe that the results of this quantitative literature review will provide a valuable tool with broad clinical applicability, potentially scalable to a large patient population. Indeed, and importantly, after initial screening—possibly using emerging technologies like actigraphy—clinicians could identify patients most likely to benefit from detailed sleep assessments, such as PSG [126,127,128]. Ultimately, this approach may enhance our diagnostic capabilities and potentially aid in predicting the course of cognitive decline, while also providing biomarkers for interventions that can be safely and effectively administered at home, as has already been suggested in various pathological contexts [129,130].

## 4. Conclusions

In conclusion, this protocol outlines a framework to investigate PSG features across neurodegenerative dementias. By systematically mapping syndrome-specific and shared alterations in sleep architecture and electrophysiology, the study could provide new insights into how sleep disruption links to neurodegeneration. Importantly, PSG-derived markers may not only improve differential diagnosis but also offer prognostic and therapeutic insights, supporting the integration of sleep metrics into precision medicine strategies. Ultimately, this work aims to develop non-invasive, scalable, and clinically meaningful biomarkers that can improve patient classification and inform personalized interventions in dementia care.

## Data Availability

Since this manuscript outlines a protocol for a systematic review and network meta-analysis, no datasets have been produced or analyzed yet. Further inquiries can be directed to the corresponding authors.

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
