# Peer review of "Sleep Disturbances Across Dementias and Cognitive Decline: Study Protocol for a Systematic Review and Network Meta-Analysis of Polysomnographic Findings"

_jcm, 2025, doi:10.3390/jcm14207437_

Round 1
Reviewer 1 Report
Comments and Suggestions for Authors
INTRODUCTION
The introduction is well-structured; however, I have a few suggestions.
Rows 76-77: The correct acronym for High-Density EEG is HD-EEG. qEEG refers to the quantitative analyses conducted on the recorded electroencephalography data. I suggest the authors modify the sentence.
Rows 121-139: I suggest the authors expand the paragraph on the role of PSG. Given that the authors emphasize the role of quantitative EEG (qEEG) metrics, it would be beneficial to provide a brief description of the neurodegenerative changes concerning slow waves and spindles.
MATERIALS AND METHODS
Row 228: I suggest the authors add the website or the reference of ‘Rayyan’.
Rows 249-252: Could the authors please provide a rationale for excluding the characteristic sleep graphoelements (slow waves, spindles, K-complexes) from the meta-analysis, considering that their quantitative metrics focused exclusively on spectral power?
RESULTS
The findings were entirely missing. Please add results, tables, and figures.
DISCUSSION
While well-written, the discussion section currently reads more like a conclusion, as it is significantly shorter than the Introduction section. Given the depth of the analyses performed, I advise the authors to discuss the findings more comprehensively
Author Response
INTRODUCTION
The introduction is well-structured; however, I have a few suggestions.
Rows 76-77: The correct acronym for High-Density EEG is HD-EEG. qEEG refers to the quantitative analyses conducted on the recorded electroencephalography data. I suggest the authors modify the sentence.
First, we thank the Reviewer for taking the time to review our manuscript and for providing valuable suggestions to improve our work. Regarding the distinction between High-Density EEG (HD-EEG) and quantitative EEG (qEEG), we appreciate the Reviewer for highlighting this important point and giving us the opportunity to clarify our rationale. Indeed, we agree that the correct acronym for High-Density EEG is HD-EEG, and we acknowledge that qEEG refers specifically to the quantitative analyses performed on EEG data. However, please note that, in our manuscript, the reference to HD-EEG was intended only in a general sense. Consistently, our study will primarily focus on qEEG analyses rather than on HD-EEG recordings. Therefore, we revised our previous statement to prevent confusion and more accurately represent the methodological focus of our work. Specifically, please see the Introduction section (lines 75-78):
“Furthermore, promising neurophysiological approaches, such as high-density electroencephalography (HD-EEG) recordings, require advanced acquisition systems and complex analytical pipelines, making them difficult to implement in routine clinical settings.”
Furthermore, in the Methods section, to clarify this distinction even more, we revised our previous sentence as follows (please see subsection 2.1.5 Types of Outcomes, lines 224-227):
“Finally, when available, we will also extract quantitative electroencephalography (qEEG) metrics, which involve the quantitative analysis of EEG signals using advanced mathematical and statistical methods, to better characterize sleep features”.
We hope this clarification addresses the Reviewer’s concern and, at the same time, improves the overall scientific and methodological quality of our work. We are sincerely grateful to the Reviewer for pointing out this important aspect.
References:
- Formica, C.; Gjonaj, E.; Bonanno, L.; Quercia, A.; Cartella, E.; Romeo, L.; Quartarone, A.; Marino, S.; De Salvo, S. The Role of High-Density EEG in Diagnosis and Prognosis of Neurological Diseases: A Systematic Review. Clin. Neurophysiol. Off. J. Int. Fed. Clin. Neurophysiol. 2025, 174, 37–47, doi:10.1016/j.clinph.2025.03.026.
- Rezaei, S.; Asadirad, F.; Motamedi, A.; Kamran, M.; Parsa, F.; Samimi, H.; Ghannadikhosh, P.; Zahmatyar, M.; Hosse-inzadeh, S.A.; Arabi, H. Future of Alzheimer’s Detection: Advancing Diagnostic Accuracy through the Integration of qEEG and Artificial Intelligence. NeuroImage 2025, 317, 121373, doi:10.1016/j.neuroimage.2025.121373.
- Sun, H.; Ye, E.; Paixao, L.; Ganglberger, W.; Chu, C.J.; Zhang, C.; Rosand, J.; Mignot, E.; Cash, S.S.; Gozal, D.; et al. The Sleep and Wake Electroencephalogram over the Lifespan. Neurobiol. Aging 2023, 124, 60–70, doi:10.1016/j.neurobiolaging.2023.01.006.
- Ye, E.; Sun, H.; Leone, M.J.; Paixao, L.; Thomas, R.J.; Lam, A.D.; Westover, M.B. Association of Sleep Electroen-cephalography-Based Brain Age Index With Dementia. JAMA Netw. Open 2020, 3, e2017357, doi:10.1001/jamanetworkopen.2020.17357.
- Hussain, I.; Hossain, M.A.; Jany, R.; Bari, M.A.; Uddin, M.; Kamal, A.R.M.; Ku, Y.; Kim, J.-S. Quantitative Evaluation of EEG-Biomarkers for Prediction of Sleep Stages. Sensors 2022, 22, 3079, doi:10.3390/s22083079.
Rows 121-139: I suggest the authors expand the paragraph on the role of PSG. Given that the authors emphasize the role of quantitative EEG (qEEG) metrics, it would be beneficial to provide a brief description of the neurodegenerative changes concerning slow waves and spindles.
We sincerely thank the Reviewer for highlighting this important point, and we agree on the importance of better exploring these essential aspects. Accordingly, we added the following clarification to our Introduction to provide the reader with a general but informative background of common alterations in the context of neurodegenerative dementias (please see the Introduction section, lines 124-128):
“In particular, changes in key features of physiological NREM sleep, such as reductions in slow-wave activity and alterations in spindle density and morphology, have been consistently reported across various neurodegenerative conditions, reflecting the early dysfunction of thalamocortical and cortical networks that are critically involved in memory consolidation and synaptic homeostasis”
References:
- Hanert, A., Schönfeld, R., Weber, F. D., Nowak, A., Döhring, J., Philippen, S., Granert, O., Burgalossi, A., Born, J., Berg, D., Göder, R., Häussermann, P., & Bartsch, T. (2024). Reduced overnight memory consolidation and associated alterations in sleep spindles and slow oscillations in early Alzheimer's disease. Neurobiology of disease, 190, 106378. https://doi.org/10.1016/j.nbd.2023.106378
- Helfrich, R. F., Mander, B. A., Jagust, W. J., Knight, R. T., & Walker, M. P. (2018). Old Brains Come Uncoupled in Sleep: Slow Wave-Spindle Synchrony, Brain Atrophy, and Forgetting. Neuron, 97(1), 221–230.e4. https://doi.org/10.1016/j.neuron.2017.11.020
MATERIALS AND METHODS
Row 228: I suggest the authors add the website or the reference of ‘Rayyan’.
We thank the Reviewer for highlighting this important point, and we fully agree on the importance of providing this relevant information. Therefore, we added the reference to the website (please see subsection 2.3 Study Screening and Selection, line 241).
Rows 249-252: Could the authors please provide a rationale for excluding the characteristic sleep graphoelements (slow waves, spindles, K-complexes) from the meta-analysis, considering that their quantitative metrics focused exclusively on spectral power?
We thank the Reviewer for this valuable comment. We fully agree that sleep graphoelements such as slow waves, spindles, and K-complexes represent relevant neurophysiological markers in this context. Our decision not to include them in the initial search strategy was driven by practical considerations, not a theoretical reason: our main goal was to compare PSG profiles across different dementia syndromes, and even within this narrower focus, the existing literature is already quite limited. For this reason, aside from specific sleep indexes, we focused on spectral power analyses, which are the most consistently reported metrics. Nevertheless, we recognize their importance for our meta-analysis and acknowledge the Reviewer’s point. Thus, we will include these graphoelements in our review whenever such data are available. Accordingly, we added the following specification to our manuscript (please see subsection 2.4 Data Extraction, lines 267-269):
“Finally, whenever reported, we will also consider characteristic sleep graphoelements (e.g., spindles, K-complexes), given their well-established relevance to sleep physiology and dementia research.”
Moreover, we also added the following clarification about outcomes of interest (please see subsection 2.1.5 Types of Outcomes, lines 206-208):
“Moreover, we will also examine characteristic sleep graphoelements (e.g., slow waves, spindles, K-complexes) whenever they are reported.”
RESULTS
The findings were entirely missing. Please add results, tables, and figures.
We appreciate the Reviewer's comment and the chance to clarify. Please note that we describe a study protocol for a systematic review and network meta-analysis. Therefore, according to the scientific methodology for this type of research, the processes of literature search, screening, and data extraction have not yet been carried out. Therefore, it is not possible to provide results, tables, or figures at this point. The rationale for publishing protocols lies precisely in enhancing methodological transparency and reducing the risk of bias, selective reporting, or post-hoc modifications once results are known. As widely recommended in the field, protocols should detail the planned objectives, eligibility criteria, and analytic strategy; results will only be reported in the subsequent systematic review and meta-analysis once the study is completed. To further clarify, please see several references to registered and published protocols which, indeed, do not include a results section:
- Paradies, Y., Priest, N., Ben, J. et al. Racism as a determinant of health: a protocol for conducting a systematic review and meta-analysis. Syst Rev 2, 85 (2013). https://doi.org/10.1186/2046-4053-2-85
- Cheloni R, Gandolfi SA, Signorelli C, et alGlobal prevalence of diabetic retinopathy: protocol for a systematic review and meta-analysisBMJ Open 2019;9:e022188. doi: 10.1136/bmjopen-2018-022188
- Jones, N.R., Roalfe, A., Adoki, I. et al. Survival of patients with chronic heart failure in the community: a systematic review and meta-analysis protocol. Syst Rev 7, 151 (2018). https://doi.org/10.1186/s13643-018-0810-x
- Kumarasamy, Chellan B.Techa; Sabarimurugan, Shanthi PhDb; Madurantakam, Royam Madhav MScb; Lakhotiya, Kartik B.Techb; Samiappan, Suja PhDc; Baxi, Siddhratha MBBS, FRANZCR, GAICDd; Nachimuthu, Ramesh PhDb; Gothandam, Kodiveri Muthukaliannan PhDb; Jayaraj, Rama MVSc, PhD, GCTLHE, MPHe,∗. Prognostic significance of blood inflammatory biomarkers NLR, PLR, and LMR in cancer—A protocol for systematic review and meta-analysis. Medicine 98(24):p e14834, June 2019. | DOI: 10.1097/MD.0000000000014834
- Gebremedhin AT, Regan AK, Malacova E, et alEffects of interpregnancy interval on pregnancy complications: protocol for systematic review and meta-analysisBMJ Open 2018;8:e025008. doi: 10.1136/bmjopen-2018-025008
We hope this clarifies an important aspect of our publication type, and we thank the Reviewer for giving us this opportunity.
DISCUSSION
While well-written, the discussion section currently reads more like a conclusion, as it is significantly shorter than the Introduction section. Given the depth of the analyses performed, I advise the authors to discuss the findings more comprehensively.
Once again, we thank the Reviewer for this relevant comment. As noted above, the present manuscript outlines a study protocol; therefore, no original results or analysis outputs are available for discussion. The current discussion is therefore grounded in solid evidence from existing literature and hypotheses about the potential significance of the results from our planned analyses. We intended to highlight how this review may contribute to clarifying the role of PSG features in neurodegenerative dementias, rather than to draw premature conclusions. Nevertheless, we appreciate the Reviewer’s suggestion and have expanded the discussion to more clearly articulate the expected contribution of this work. Specifically, please see the Discussion section (lines 353-360, lines 366-370, and lines 375-378):
“Thus, it is hoped that this work will reveal whether specific PSG signatures differ systematically across dementia syndromes. Such findings would provide essential insights into the pathophysiological mechanisms linking neurodegeneration to sleep disruption, such as highlighting the selective vulnerability of thalamocortical or brainstem circuits in different dementia subtypes. Importantly, by combining sleep features with current neurodegeneration models, we can explore whether specific PSG features represent downstream markers of widespread network issues or early factors that drive disease progression.”
“This approach could foster the integration of sleep metrics into precision medicine frameworks, paving the way for early detection, patient-tailored management, and the development of novel therapeutic targets. Notably, PSG-derived biomarkers may hold prognostic value, enabling clinicians to stratify patients based on their risk of rapid cognitive decline or responsiveness to targeted interventions.”
“Ultimately, this approach may enhance our diagnostic capabilities and potentially aid in predicting the course of cognitive decline, while also providing biomarkers for interventions that can be safely and effectively administered at home, as has already been suggested in various pathological contexts.”
Additionally, we added a new Conclusions section to give the reader a final overview of this work and its aims (please see section 4. Conclusions, lines 380-388):
“In conclusion, this protocol outlines a framework to investigate PSG features across neurodegenerative dementias. By systematically mapping syndrome-specific and shared alterations in sleep architecture and electrophysiology, the study could provide new insights into how sleep disruption links to neurodegeneration. Importantly, PSG-derived markers may not only improve differential diagnosis but also offer prognostic and therapeutic insights, supporting the integration of sleep metrics into precision medicine strategies. Ultimately, this work aims to develop non-invasive, scalable, and clinically meaningful biomarkers that can improve patient classification and inform personalized interventions in dementia care.”
References:
- Ahmed, R.M.; Devenney, E.M.; Irish, M.; Ittner, A.; Naismith, S.; Ittner, L.M.; Rohrer, J.D.; Halliday, G.M.; Eisen, A.; Hodges, J.R.; et al. Neuronal Network Disintegration: Common Pathways Linking Neurodegenerative Diseases. J. Neurol. Neurosurg. Psychiatry 2016, 87, 1234–1241, doi:10.1136/jnnp-2014-308350.
- Musiek, E.S.; Holtzman, D.M. Mechanisms Linking Circadian Clocks, Sleep, and Neurodegeneration. Science 2016, 354, 1004–1008, doi:10.1126/science.aah4968.
- Satorres, E.; Escudero Torrella, J.; Real, E.; Pitarque, A.; Delhom, I.; Melendez, J.C. Home-Based Transcranial Direct Current Stimulation in Mild Neurocognitive Disorder Due to Possible Alzheimer’s Disease. A Randomised, Single-Blind, Con-trolled-Placebo Study. Front. Psychol. 2023, 13, doi:10.3389/fpsyg.2022.1071737.
- Antonioni, A.; Baroni, A.; Fregna, G.; Ahmed, I.; Straudi, S. The Effectiveness of Home-Based Transcranial Direct Current Stimulation on Chronic Pain: A Systematic Review and Meta-Analysis. Digit. Health 2024, 10, 20552076241292677, doi:10.1177/20552076241292677.
Additionally, a native speaker with expertise in the subject has extensively revised the text to enhance the quality of scientific English. We sincerely thank the Reviewer once again for these valuable advices and for helping us improve the scientific rigor of our manuscript.
Reviewer 2 Report
Comments and Suggestions for Authors
The manuscript is well-written and addresses an important gap in the literature by systematically synthesizing polysomnographic (PSG) findings across the dementia spectrum. The rationale is clear, the methods are rigorous, and the potential clinical impact is high.
Major strengths include adherence to PRISMA-NMA guidelines, PROSPERO registration, comprehensive inclusion/exclusion criteria, well-defined PSG and cognitive/behavioral outcomes, and a detailed plan for sensitivity/subgroup analyses.
Points for improvement:
- Please provide at least one full database search string (PubMed) in the appendix/supplementary material for transparency.
- Clarify how studies reporting medians/IQRs will be converted to means/SDs (e.g., Wan et al. or Luo et al. methods).
- Specify how heterogeneity from comorbidities, sex, or age distributions will be managed (e.g., subgroup or meta-regression).
- Explicitly mention whether standardized diagnostic frameworks (Petersen for MCI, Jessen for SCI) are required.
- Consider whether GRADE certainty of evidence assessment will be applied to the final synthesis.
- A planned PRISMA flow diagram for study selection would improve clarity.
Author Response
The manuscript is well-written and addresses an important gap in the literature by systematically synthesizing polysomnographic (PSG) findings across the dementia spectrum. The rationale is clear, the methods are rigorous, and the potential clinical impact is high.
Major strengths include adherence to PRISMA-NMA guidelines, PROSPERO registration, comprehensive inclusion/exclusion criteria, well-defined PSG and cognitive/behavioral outcomes, and a detailed plan for sensitivity/subgroup analyses.
Points for improvement:
- Please provide at least one full database search string (PubMed) in the appendix/supplementary material for transparency.
First, we thank the Reviewer for taking the time to review our manuscript, for this positive feedback, and for offering valuable suggestions to enhance our work. We entirely agree and thank the Reviewer for highlighting this relevant point. Accordingly, please acknowledge that the previous paragraph on the search strategy "The MEDLINE (via PubMed), Scopus, EBSCO, and Web of Science databases will be searched for studies reporting nocturnal PSG evaluations of individuals with SCI, MCI, AD, FTD, LBD, or PDD, compared to HS. Medical Subject Headings (MeSH) terms and natural language terms will be used together in the search strategy tailored for each specific database by a research team expert" has been rephrased as follows (please see subsection 2.2 Search Strategy, lines 230-238):
"A bibliographic search will be performed across different databases (MedLine, Scopus, EBSCO and Web of Science), with no date restriction. Various combinations of the following search terms related to eight main domains will be used, after adjustment for each database: "polysomnografy OR PSG"(title/abstract) AND "sleep"(title/abstract) AND "Alzheimer's Disease OR AD"(title/abstract) or "subjective cognitive impairment OR SCI" (title/abstract) or "mild cognitive impairment OR MCI"(title/abstract) or "fronto-temporal dementia OR FTD"(title/abstract) or "Lewy body dementia OR LBD"(title/abstract) or "Parkinson's disease dementia OR PDD"(title/abstract). The reference lists of reviews and retrieved articles will be also searched for additional pertinent papers".
- Clarify how studies reporting medians/IQRs will be converted to means/SDs (e.g., Wan et al. or Luo et al. methods).
We agree that this issue deserves clarification, and apologize for not having provided details before. Accordingly, please acknowledge that we added the following lines to the Data Synthesis section (please see lines 305-308):
"In case some of the included studies express PSG parameters as medians and ranges, we will use the method described by Hozo et al to obtain the corresponding means and SDs. In case interquartile ranges (IQR) rather than ranges are reported, they will be divided by 1.35 to obtain the equivalent SD".
Please also acknowledge that the following references were added:
- Hozo SP, Djulbegovic B, Hozo I: Estimating themean and variance from the median, range, and the size of a sample. BMC Med Res Methodol 2005, 5:13.
- Higgins JPT, Green, S.,. Cochrane Handbook for Systematic Reviews of Interventions: The Cochrane Collaboration; 2011. Available from: www.cochrane-handbook.org.
- Specify how heterogeneity from comorbidities, sex, or age distributions will be managed (e.g., subgroup or meta-regression).
We agree and thank the Reviewer for emphasizing this important aspect. Accordingly, please acknowledge that the previous lines in the subsection "2.7 - Analysis of subgroups or subsets" "Additionally, if more than 10 studies are available, a meta-regression will be performed to explore the association between the selected covariates and the outcome of interest" have been rephrased as follows (lines 324-330):
"Additionally, if sufficient data will be available, all analyses will be stratified by several demographic and clinical characteristics: (a) gender; (b) clinical status (presence vs. absence of comorbidities); (c) type of treatment (pharmacological; non-pharmacological; no treatment). Moreover, in case more than 10 studies are available, a meta-regression will be performed to explore the association between the selected covariates and the outcome of interest".
- Explicitly mention whether standardized diagnostic frameworks (Petersen for MCI, Jessen for SCI) are required.
We sincerely thank the Reviewer for highlighting this critical point, and we agree on the importance of clarifying this aspect. Indeed, we added the relevant references to the diagnostic criteria in MCI and SCI patient categories as well to enhance scientific rigour and transparency in the subsections 2.1.1 Types of Studies (please see lines 170-171) and 2.1.2 Types of Populations (please see lines 186-188):
- Petersen, R.C. Mild Cognitive Impairment. Continuum 2004, 10, 9–28, doi:10.1212/01.CON.0000293545.39683.cc.
- Jessen, F.; Amariglio, R.E.; Buckley, R.F.; Flier, W.M. van der; Han, Y.; Molinuevo, J.L.; Rabin, L.; Rentz, D.M.; Rodriguez-Gomez, O.; Saykin, A.J.; et al. The Characterisation of Subjective Cognitive Decline. Lancet Neurol. 2020, 19, 271–278, doi:10.1016/S1474-4422(19)30368-0.
- Consider whether GRADE certainty of evidence assessment will be applied to the final synthesis. A planned PRISMA flow diagram for study selection would improve clarity.
Once again, we thank the Reviewer for this insightful comment. We agree that this issue deserves clarification, and apologize for not having provided details before. Accordingly, please acknowledge that we added the following lines to subsection 2.5 - Quality assessment (lines 276-281):
"The strength of the body of evidence will be assessed through the Grading of Recommendations, Assessment, Development, and Evaluations (GRADE) certainty ratings. Using GRADE guidelines, the researchers will examine the following domains for a decrease in certainty: risk of bias, imprecision of true effects, inconsistency of effects, indirectness of outcomes, and publication bias".
Please also acknowledge that the following new references were added:
- Guyatt GH, Oxman AD, Kunz R, Vist GE, Falck-Ytter Y, Schunemann HJ. What is “quality of evidence” and why is it important to clinicians? BMJ 2008;336(7651):995–8.
- Guyatt GH, Oxman AD, Vist GE, Kunz R, Falck-Ytter Y, Alonso-Coello P, et al. GRADE: an emerging consensus on rating quality of evidence and strength of recommendations. BMJ (Clinical research ed) 2008;336(7650):924–6.
Finally, please acknowledge that we added a PRISMA flow diagram showing the study selection process at each stage, and we mentioned this issue adding the following lines to subsection 2.3 - Study screening and selection (lines 246-248): "A PRISMA flow diagram showing the number of studies included and excluded at each stage of the study selection process will be provided (Figure S1)"
Please also acknowledge that the following reference was added:
- Moher D, Liberati A, Tetzlaff J, Altman DG, The PRISMA Group. Preferred Reporting Items for Systematic Reviews and Meta-Analyses: The PRISMA Statement. PLoS Med 2009; 6(6): e1000097.
Additionally, a native speaker with expertise in the subject has extensively revised the text to enhance the quality of scientific English. We sincerely thank the Reviewer once again for these valuable advices and for helping us improve the scientific rigor of our manuscript.
Round 2
Reviewer 1 Report
Comments and Suggestions for Authors
The authors have fully addressed my requests. I would, however, like to clarify a point. Although a protocol is being presented, which does not typically require the inclusion of results, it would have been preferable to integrate the decision-making process graphs and tables, which I initially referred to as 'results’, within the main body of the manuscript. Nevertheless, I accept the decision to present them as supplementary material.